# The Monitoring of Black-Odor River by Electronic Nose with Chemometrics for pH, COD, TN, and TP

Shanshan Qiu [1,*], Pingzhi Hou [2], Jingang Huang [1], Wei Han [1] and Zhiwei Kang [3]

1   College of Materials and Environmental Engineering, Hangzhou Dianzi University, Hangzhou 310018, China; hjg@hdu.edu.cn (J.H.); hanwei1982@hdu.edu.cn (W.H.)
2   One Belt and One Road Information Research Institute, Hangzhou Dianzi University, Hangzhou 310018, China; houpingzhi@hdu.edu.cn
3   Bureau of Ecology and Environment of Qinghe City, 49 Zhujiang Road, Xingtai 054800, China; moottery@163.com
*   Correspondence: qiuss@hdu.edu.cn; Tel.: +86-15858152633

**Abstract:** Black-odor rivers are polluted urban rivers that often are black in color and emit a foul odor. They are a severe problem in aquatic systems because they can negatively impact the living conditions of residents and the functioning of ecosystems and local economies. Therefore, it is crucial to identify ways to mitigate the water quality parameters that characterize black-odor rivers. In this study, we tested the efficacy of an electronic nose (E-nose), which was inexpensive, fast, and easy to operate, for qualitative recognition analysis and quantitative parameter prediction of samples collected from the Yueliang River in Huzhou City. The E-nose sensors were cross-sensitive to the volatile compounds in black-odor water. The device recognized the samples from different river sites with 100% accuracy based on linear discriminant analysis. For water quality parameter predictions, partial least squares regression models based on E-nose signals were established, and the coefficients between the actual water quality parameters (pH, chemical oxygen demand, total nitrogen content, and total phosphorous content) and the predicted values were very high ($R^2 > 0.90$) both in the training and testing sets. These results indicate that E-nose technology can be a fast, easy-to-build, and cost-effective detection system for black-odor river monitoring.

**Keywords:** electronic nose; black-odor river; headspace gas; recognition; prediction





## 1. Introduction

Black-odor rivers are polluted urban rivers that often are black in color and emit a foul odor. They are a severe problem in aquatic systems affected by urbanization, as they negatively impact the living conditions of local residents, the functioning of ecosystems, and the quality of urban landscapes [1]. Black odorous rivers have become widespread problems since 1962 [2], and they seem to occur more frequently in less developed areas in recent years.

Some studies of black-odor rivers have focused on the mechanisms responsible for their formation, including the compounds that contribute to blackening and odor formation, developmental conditions, and biogeochemical processes. Other studies have focused on controlling or treating black-odor rivers, such as artificial aeration, sediment dredging, microbial enhancement technologies, and constructed wetlands [1]. However, reports of monitoring of black-odor rivers are rare.

Generally, studies of black-odor rivers can be categorized as either in situ or ex situ. In situ studies involve adding chemical or biological reagents directly into the water at a given site [3]. However, such studies require a relatively long process time, and conditions can be unstable. In some cases, they can even lead to the addition of new pollutants into the river if conducted improperly [4]. Ex situ studies involve removing contaminants from

black-odor rivers using chemical reagents at a remote location, but they cannot detect the water quality parameters immediately on site.

Studies of the mechanisms responsible for the formation of black-odor rivers involve the measurement of water quality parameters, such as pH, chemical oxygen demand (COD), total nitrogen (TN) content, and total phosphorus (TP) content, which are the most important contamination indicators of rivers or lakes. They are measured using traditional chemical detection methods. However, these detection methods require chemical reagents and/or laboratory equipment, so obtaining results takes time, and they are not immediately available. Moreover, techniques such as the dichromate method to measure COD, the alkaline potassium persulfate digestion-UV spectrophotometric method to measure TN, and the ammonium molybdate spectrophotometric method to measure TP are expensive, time- and labor-intensive, and require professional expertise. Therefore, alternative measurement methods are needed.

One such innovative measurement system is the electronic nose (E-nose), which is designed to mimic human olfactory perception [5]. An E-nose is a multisensory array that is sensitive to gaseous chemicals [6], is capable of detecting odors, vapors, gases, and volatile compounds, and can produce electronic responses upon exposure to headspace gas [7]. E-nose technology provides the opportunity to exploit the information contained in the headspace gas of samples in many different fields of study [8], such as agricultural and food production [9], lung cancer histological type analysis [10], mint scents identification [11] and fungal infection recognition [12]. Initially developed as an inexpensive, fast, and easy tool to detect volatile compounds, its application also may provide the required time-resolved coverage of odor emission phenomena cheaply and feasibly with respect to chemical analysis.

The application of E-nose technology to environmental detection has mainly focused on air detection, such as indoor air quality monitoring [13], gas discrimination [14], and toxic and non-toxic substance discrimination [15]. However, a few researchers have reported the use of E-nose monitoring of the aquatic environment. Such monitoring ability is urgently needed, especially for black-odor rivers. An E-nose can be used to detect the water quality parameters of black-odor water in situ, eliminate the need for chemical reagents, and is inexpensive. This technology also delivers immediate detection results, which ex situ methods cannot provide.

BOD and COD are the most important physic-chemical properties for the black-odor rivers, reflecting the contamination level when the river has become a black-odor river. We have also detected the pH and TP to enrich the experiment data and to check the E-nose could be adaptable for more quality parameters. In this study, we tested the efficacy of an E-nose device to collect data from the headspace gas of samples collected from a black-odor river. The main objectives of this study were to (1) measure water quality parameters (pH, COD, TN, and TP contents, which indicate the pollution levels of black odors river) using traditional techniques; (2) analyze the E-nose sensor behaviors during headspace gas detection of the samples, and (3) test the ability of the E-nose to recognize the black-odor river samples and to predict their water quality parameters.

## 2. Materials and Methods

### 2.1. Sample Collection and Preparation

Black odors river water samples were collected from Yueliang River (latitude 119.7° N, longitude 30.8° E) with a total area of 2.8 km² and a total length of 8.3 km, in Huzhou, Zhejiang province, in China.

Six sampling sites (named group 1 to group 6 in this study) with intervals of 1 km were set up along the river on 27 October 2019. Samples were collected using plastic bottles with a capacity of 1.5 L, which was enough for E-nose detection and conventional analytical detections. The lab temperature was controlled at $20 \pm 0.5$ °C once the samples were collected and arrived.

## 2.2. Water Quality Analysis

### 2.2.1. Total Phosphorous and Total Nitrogen

The total phosphorous (TP) and total nitrogen (TN) in the water samples were determined by strictly following the Chinese standard methods (GB 11893-89 [16] and GB 11894-89 [17]). After water samples were digested by mixed concentrated nitric and perchloric acids ($HNO_3$-$HClO_4$), the ammonium molybdenum ascorbic acid method was used to detect TP content. The concentrations of TN in water samples were analyzed using the alkaline potassium persulfate oxidation-UV spectrophotometric method.

### 2.2.2. pH and Chemical Oxygen Demand

pH and chemical oxygen demand (COD) in the water samples were determined by strictly following the Chinese standard methods (GB 6980-1986 [18] and GB 11914-89 [19]). pH value was detected by an Eh-pH meter (PHS-25, Shanghai Precision and Scientific Instrument CO.LTD, China) directly. The value of COD was detected using the dichromate method.

Water quality detections were all accomplished on the sample collection day. Three replications were performed for analysis, and the mean values and mean square deviations were presented.

## 2.3. Electronic Nose Detection

Electronic nose (E-nose), as the name suggests, is a detection instrument to mimic the human nose, which comprises an array of electronic chemical sensors capable of recognizing simple or complex odors, a sampling channel, and a data acquisition system [20]. E-nose Unlike other detection instruments, E-nose does not test specific ingredients but gives the sample headspace gas complementary information [21]. When the sensors interact with the headspace gas delivered by the sampling channel, the E-nose system processes the sensor electronic signals to digital data, which is usually called the fingerprint data of the sample [22].

In this study, a commercial PEN2 E-nose (Airsense Analytics, GmBH, Schwerin, Germany) was applied to detect the headspace gas of black-odor river samples. In this E-nose device, metal oxide sensors are chosen as the core part. They can turn headspace gas measurements, such as methyl groups, hydrogen sulfide, propane, and odor chemicals into signals.

After a training step, data mining methods can identify or predict information about single compounds or mixtures of gases. Table 1 lists all sensors used, their main applications, currently known or specified reactions, and detection limits.

E-nose detects the headspace gas of samples, so the only sample treatment is headspace gas gathering without any sample pretreatment. Thus, a 10 mL sample of black-odor river water was placed in a 500 mL beaker, sealed with plastic wrap, and the beaker was kept still for 30 min to allow the generation of headspace gas. Before measurement, a hole was made in the plastic wrap to provide a steady stream of gas for the E-nose to detect. The headspace gas path and the sensor chamber in the E-nose device were cleaned with clean air within 60 s after taking a measurement. The flow rate of headspace gas was 200 mL min$^{-1}$, and during E-nose detection, one signal per second was collected. The measurement duration was 75 s to ensure stable sensor signals. The E-nose signal is expressed as G/G0, where G and G0 represent the resistance of a sensor in sample headspace gas and clean air, respectively.

For E-nose detection, 10 mL of water sample was prepared for each sample, and 27 samples of each group were prepared for each sampling site. The water sample was put into a 500 mL glass beaker, and the beaker was sealed with plastic wrap. The sample was standstill for 30 min for headspace gas generation and gas balance. The E-nose detection was accomplished on the sample collection day. Contamination scales suitable to compare data obtained from E-nose and from quality properties were (7.1~8.0), COD (48~139 mg/L), TN (17.1~32.7 mg/L), TP (1.00~2.48 mg/L). 2.4. Data analysis.

**Table 1.** Sensors used and their main applications in the E-nose (PEN2, Airsense Analytics, Germany).

| No. | Sensor Name | General Description | Reference |
|---|---|---|---|
| S1 | W1C | Aromatic compounds | Toluene, 0.1 g/kg |
| S2 | W5S | Very sensitive with negative signal, broad range sensitivity, react on nitrogen oxides | $NO_2$, $1 \times 10^{-3}$ g/kg |
| S3 | W3C | Very sensitive with aromatic compounds | Benzene, $1 \times 10^{-2}$ g/kg |
| S4 | W6S | Mainly hydrogen, selectively, (breath gases) | $H_2$, 0.1 g/kg |
| S5 | W5C | Alkanes, aromatic compounds, less polar compounds | Propane, $1 \times 10^{-3}$ g/kg |
| S6 | W1S | Sensitive to methane (environment). Broad range, similar to S8; | $CH_3$, 0.1 g/kg |
| S7 | W1W | Reacts on sulfur compounds, or sensitive to many terpenes and sulfur organic compounds; | $H_2S$, $1 \times 10^{-4}$ g/kg |
| S8 | W2S | Detects alcohol's, partially aromatic compounds, broad range | CO, 0.1 g/kg |
| S9 | W2W | Aromatics compounds, sulfur organic compounds | $H_2S$, $1 \times 10^{-3}$ g/kg |
| S10 | W3S | Reacts on high concentrations >0.1 g/kg, sometime very selective (methane) | $CH_3$, 0.1 g/kg |

### 2.3.1. Linear Discriminant Analysis

Linear discriminant analysis (LDA) combines Fisher's linear discriminant, analysis of variance (ANOVA), and regression analysis to find a set of linear equations that can separate two or more classes of samples [23].

The five simple steps listed below are required is to perform a general LDA analysis:

1.  D-dimensional mean vectors are computed from the original dataset, which includes different classes (6 groups in this study);
2.  Compute the between-class-matrix and within-class-matrix;
3.  Compute the eigenvectors and corresponding eigenvalues from between-class-matrix and within-class-matrix;
4.  Sort the eigenvectors by decreasing eigenvalues and choose k eigenvectors with the largest eigenvalues.
5.  Use the eigenvector matrix to transform the samples onto the new subspace. The original data can be projected to minimize the variance in the same group and maximize the distance in the different groups.

According to the previous definition, LDA was used here to identify and choose samples of a group that are highly likely belong to some state of river samples.

### 2.3.2. Partial Least Squares Regression

Partial least squares regression (PLSR) provides the optimal linear model in terms of predictivity by a linear transition from a large number of original data to a small number of orthogonal vectors [24]. PLSR is especially useful when the predictors exhibit multicollinearity or there are more predictors than cases.

The first step of PLSR is to create two matrices: X-matrix and Y-matrix, where the X-matrix is composed of E-nose sensor signals and Y-matrix is composed of water quality parameters. These matrices are then column-centered and normalized. The five simple steps listed below are to perform a general PLSR analysis.

1.  $w \propto E^T u$ (estimate X weights);
2.  $t \propto Ew$ (estimate X factor scores);
3.  $c \propto F^T t$ (estimate Y weights);

4.     u = Fc (estimate Y scores).

$R^2$ and root mean squared error (RMSE) is used here to estimate the accuracy of the PLSR models.

### 2.3.3. Analysis of Variance–Partial Least Squares Regression

Analysis of Variance (ANOVA) is a prominent method to analyze the data structure by decomposing the total data into different sources of variance, and PLSR, which can elucidate predictive links between the sensory and chemical data. ANOVA-PLS combines variance decomposition to extract different effects and subsequent statistical analysis and slightly differs from ANOVA. Different combinations of effects are used to determine the relationship between the data types rather than single effects [25]. In the ANOVA-PLSR model, the X-matrix was set as E-nose sensor signals and the Y-matrix as the water quality parameters. Regression coefficients were analyzed by jack-knifing based on cross-validation and stability plots, thereby deriving significance indications for the relationships determined in the quantitative ANOVA-PLSR. To observe effects caused by random errors, the significance ($p < 0.05$) of the variable relationship in the X-matrix and Y-matrix was determined.

### 2.4. Software

Descriptive statistical analysis providing means, standard deviations, correlation and percentile distributions was performed to gain an overview of the sensory terms and chemical measurements. The sensory data utilized was the raw data averaged over the assessor and replicate. LDA was performed by SPSS 16.0 (IBM, Chicago, IL, USA), PLSR was processed by MATLAB 2012a software (MathWorks, Natick, MA, USA), ANOVA-PLSR used the Unscrambler software version 10.3 (Camo, Oslo, Norway).

## 3. Results and Discussion

### 3.1. The Water Quality Parameters of Black Odors River Based on Conventional Analytical Methods

To evaluate the black-odor river water quality at the six different sites in the Yueliang River, pH, COD, and TN and TP contents of the samples were measured (Table 2). The pH values of the samples from the six sites ranged from 7.2 to 7.5 and did not differ significantly. COD is an indicator of the amount of oxygen consumed by reactions in a water sample. It provides information about the potential impact of oxygen consumption on the oxygen content of the water body. The values of COD varied from 50 to 115 mg/L and differed significantly among sites. The differences in COD among sites suggest that the sites differed in their degree of contamination, and pollution at all sites mainly was due to resident activities because no industrial wastewater was poured into the Yueliang River. TN, which is the sum of total Kjeldahl nitrogen (ammonia, organic and reduced nitrogen) and nitrate-nitrite, measures the eutrophication of the body of water. The values of TN varied from 20 to 30 mg/L and differed among sites, according to Table 2. TP is the sum of all phosphorus compounds, including orthophosphate ($PO_4^{3-}$), and forms present in organic compounds. TP content provides information about water quality and trophic state. The value of TP varied from 1.8 to 2.2 and did not differ significantly among the six sampling sites. Overall, pH and TP content were stable, whereas COD and TN differed significantly among sites.

To describe correlation among water quality parameters, we used Pearson correlation matrix analysis to evaluate the values of pH, COD, and TN and TN contents (in Figure 1). In the figure, values close to 0 and 1 correspond to the low and high correlation of water quality parameters according to color depth, respectively. The values of pH, COD, and TN and TN contents showed a very low correlation with each other in Figure 1. The data distribution showed no significant pattern along the stretch of the Yueliang River sampled in this study. These results suggest that the pollution level in this area had no specific rules to follow. The differences in parameter values among sites might indicate that multi-point

dynamic monitoring is required to assess the water quality of black-odor rivers. The E-nose system is a promising option.

**Table 2.** The pH, COD, TN, and TP values in different samples. Values within the same column followed by different letters are significantly different according to the Tukey's HSD test ($p < 0.05$).

| No. | pH | COD (mg/L) | TN (mg/L) | TP (mg/L) |
|---|---|---|---|---|
| Sample 1 | 7.7 ± 0.3 [a] | 115 ± 4 [b] | 29.5 ± 3.2 [a] | 1.81 ± 0.31 [a] |
| Sample 2 | 7.2 ± 0.1 [a] | 100 ± 5 [c] | 26.1 ± 2.5 [ab] | 1.62 ± 0.23 [a] |
| Sample 3 | 7.3 ± 0.2 [a] | 82 ± 4 [d] | 20.2 ± 3.1 [b] | 1.51 ± 0.22 [a] |
| Sample 4 | 7.4 ± 0.1 [a] | 50 ± 2 [e] | 25.0 ± 1.6 [ab] | 1.34 ± 0.34 [a] |
| Sample 5 | 7.5 ± 0.2 [a] | 134 ± 5 [a] | 27.0 ± 2.1 [ab] | 1.69 ± 0.42 [a] |
| Sample 6 | 7.4 ± 0.1 [a] | 73 ± 3 [d] | 21.0 ± 2.5 [b] | 2.13 ± 0.35 [a] |

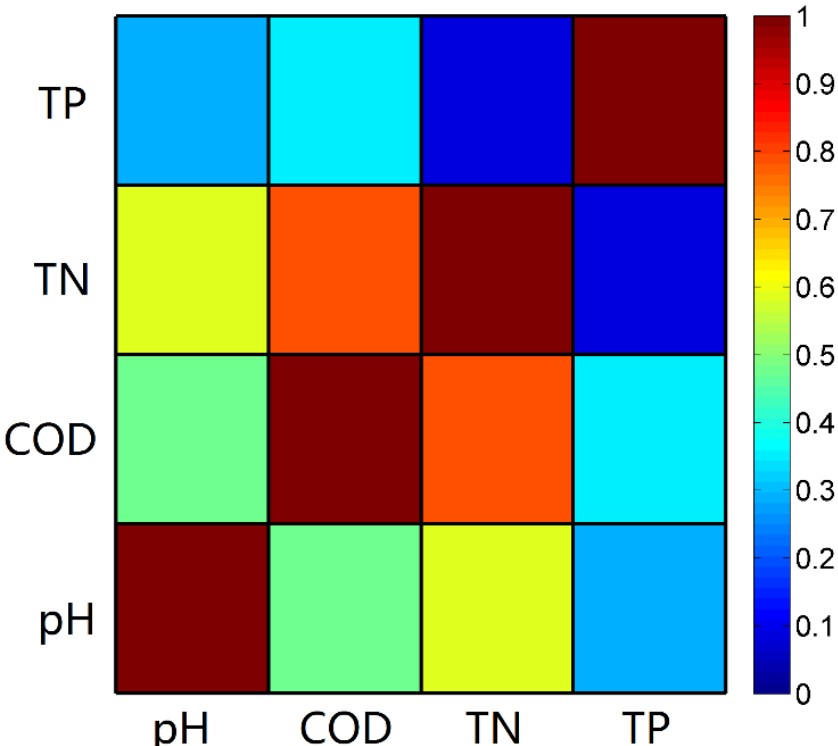

**Figure 1.** Visualization of the Pearson correlation matrix among the four quality parameters (pH, COD, TN, and TP). Values close to 0 and 1 correspond to low and high correlation, respectively.

*3.2. Response Curves of E-Nose Sensors for Black-Odor River Water Samples*

Figure 2a shows a typical response curve for the 10 E-nose sensors, with the signals expressed as G/G0. Each curve represents a sensor change during sample measurement. All the sensors reached a dynamic balance after 20 s, meaning that the E-nose device could achieve complete detection of black-odor river water samples in less than 30 s (Figure 2a). However, to ensure that we obtained stable signals, E-nose signals were collected until the 70th second in the original data. Figure 2b shows the mean values of 27 samples from the six sample sites at the 70th. The results indicate that sensors S2 and S9 were most sensitive to the black-odor river water headspace gas. Sensors S1, S3, S5, S6, and S7 also showed some sensitivity to the headspace gas, but S4 and S10 stayed almost still during the entire E-nose experiment.

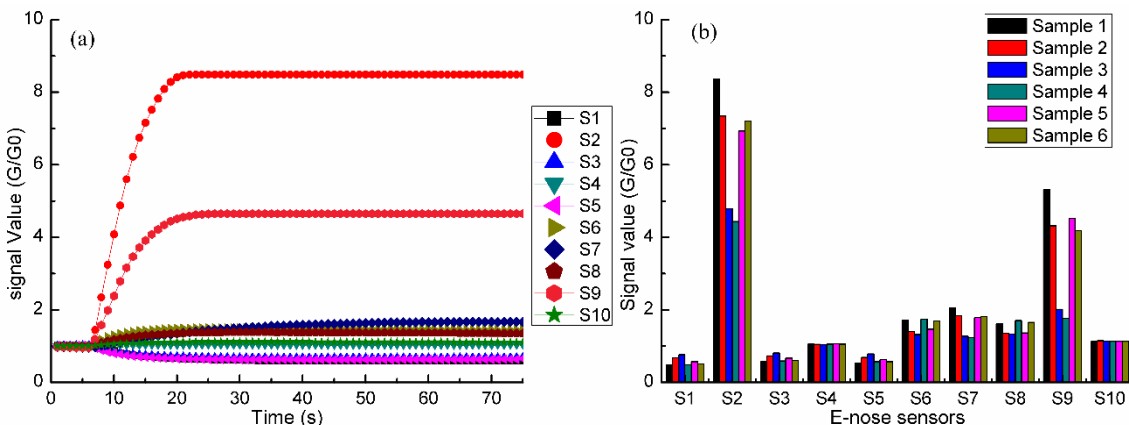

**Figure 2.** E-nose responses for samples: (**a**) were typical responses of E-nose sensors for black odors river; (**b**) were mean values of ten sensors for black odors river.

To describe the multicollinearity among sensor signals, we used Pearson correlation matrix analysis to evaluate the data from the 10 E-nose sensors, and we detected a high degree of multicollinearity (Figure 3). In the figure, values close to 0 and 1 correspond to the low and high correlation of sensor signals according to color depth, respectively. Sensors S4 and S10 showed the least correlation with other sensors because they were unreactive during the experiment. Sensors S1, S2, S3, S5, S6, S7, S8, and S10 showed different correlation degrees. Multicollinearity among independent variables hinders the regression analysis, making it difficult to precisely estimate the distinct effect of certain independent variables on a dependent variable. To reduce the risk of multicollinearity, we used partial least squares regression (PLSR) analysis to establish regression models between the E-nose signals and the measured values of water quality of black-odor river water samples.

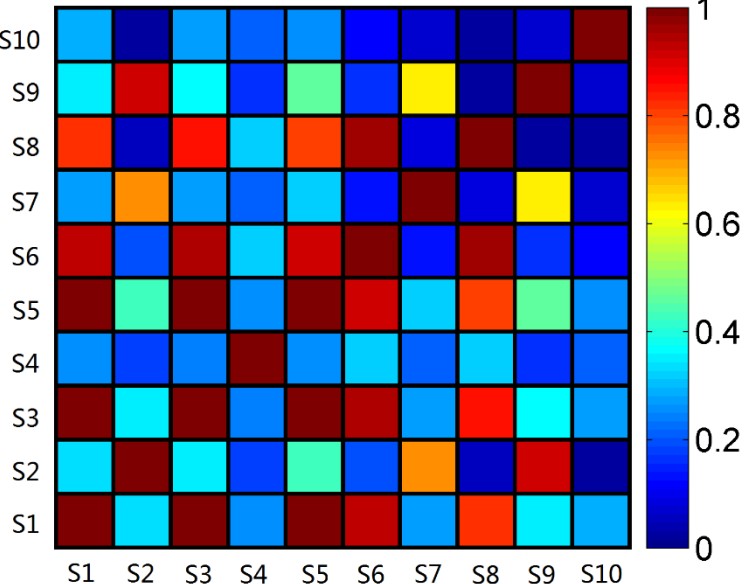

**Figure 3.** Visualization of the Pearson correlation matrix among the ten E-nose sensors evaluated over the entire dataset. Values close to 0 and 1 correspond to the low and high correlation of sensor signals, respectively.

### 3.3. Analysis of Black-Odor River Water Samples by the E-Nose System

3.3.1. Recognition of Black-Odor River Water Samples Based on Linear Discriminant Analysis (LDA)

To train the E-nose to recognize black-odor river water samples, we conducted a stepwise LDA analysis to process the E-nose data. Before LDA analysis, the signals were normalized by zero-mean normalization. Thus, the mean of those data was 0, and the covariance was 1. Then, Wilks' lambda method was applied to decide which variables should be included in the LDA model, with the usual F (0.05) probabilities for a variable to be included and F (0.10) for it to be removed. To avoid over-optimistic data modulation, we also applied a leaving-one-out cross-validation procedure.

The 10 E-nose sensor signals at the 70th second were input into the LDA model at first. After the Wilks' lambda method procedure, ten variables were included in the LDA model with five discriminant functions. The first two discriminant functions explained 62.9% and 27.2% of the total variance of the E-nose data and showed that the distribution of the samples from the six sites was clear with no sample overlapping (Figure 4a). In Figure 4b, the third discriminant function explained 6.4% of the total variance, and the distribution of the six groups also was clear with no sample overlapping (Figure 4b). These results indicate that the E-nose recognized all the samples well. Thus, 100% of original grouped cases were correctly classified, and 100% of cross-validated grouped cases were correctly classified.

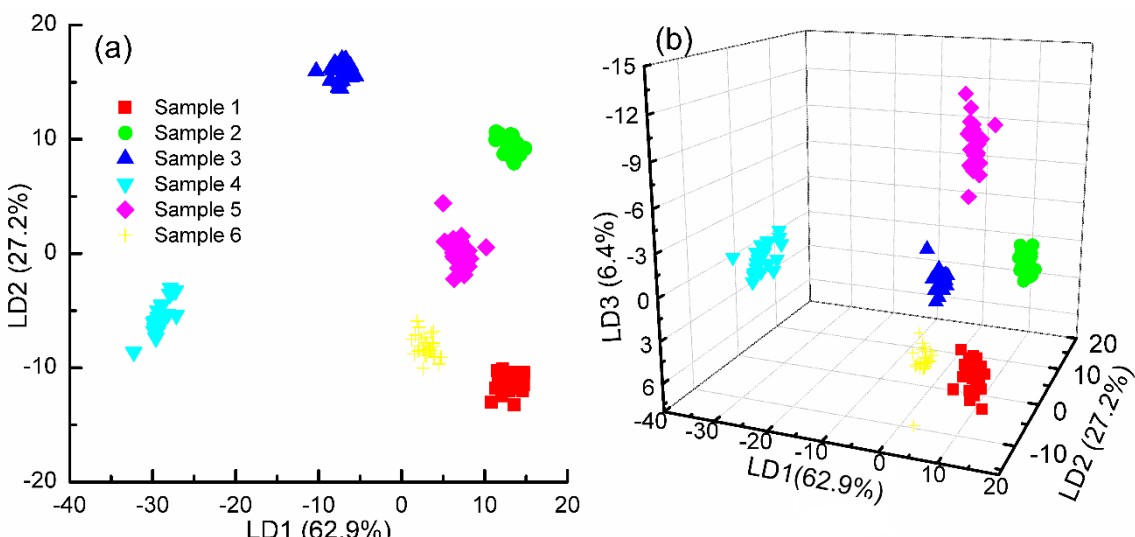

**Figure 4.** LDA analysis of black odors river samples resulted from E-nose: (**a**) Pearson correlation matrix for juices with the two-dimensional dataset; (**b**) Pearson correlation matrix for the three-dimensional dataset.

3.3.2. Correlation between Water Quality Parameters and E-Nose Sensor Signals

To analyze the correlation between the water quality parameters and E-nose sensor signals, ANOVA-PLSR was applied. The X-matrix consisted of the E-nose sensor signals at the 70th second, and the Y-matrix consisted of the water quality parameter values. The derived ANOVA-PLSR model explained 98% of the variance for the X-matrix and 94% of the variance for the Y-matrix in two principal components (PCs) in Figure 5. The two ellipses represent $R^2$ = 50 and 100%, respectively; values inside of the small ellipse were poorly correlated, whereas variables between the small and the big ellipses were correlated intensively. Some variables (such as S2 and S9) from the X-matrix showed a significant and positive correlation with a variable from the Y-matrix (COD), and other X-matrix variables showed some level of correlation with variables from the Y-matrix. A correlation with a specific compound may be detected because the E-nose sensors are cross-sensitive to all materials in the headspace gas.

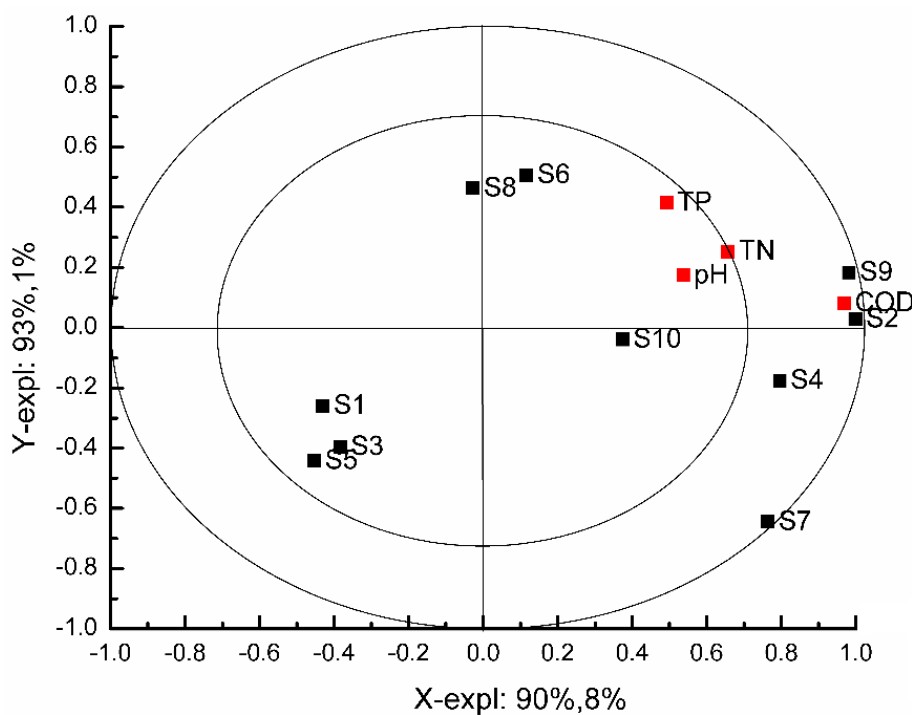

**Figure 5.** ANOVA-PLSR correlation loadings plot for PC1 versus PC2 of E-nose and river parameters of black odors river. The model was derived from the E-nose sensor signals the X-matrix (black square) and river parameters as the Y-matrix (red square). The small and the big ellipses represent $R^2$ = 50 and 100%, respectively.

### 3.3.3. Prediction of Quality Parameters of Black-Odor River Samples Based on PLSR

To establish the prediction model for water quality parameters, we performed PLSR analysis for the E-nose data. The PLSR regression models to compare E-nose sensor signals with the values of water quality parameters obtained by conventional analytical methods were generated using the leave-one-out technique. To assess the regression models, data were divided into 120 samples for model training and 42 samples for model testing (or validation). With this method, larger accuracy rates for both the training set and testing set indicate a better model.

Table 3 lists the training and testing results. For PLSR prediction models, the coefficients for the comparison of the actual water quality parameters (pH, COD, and TN and TP contents) and the predicted values were very high ($R^2$ > 0.90) for both the training and testing sets. These results indicate that the E-nose device assessed the black-odor river water's parameters based on headspace gas samples. The distribution of actual and predicted values for the 162 samples shows close relationships between quality parameters obtained using conventional analytical methods and those estimated by the PLSR models based on E-nose data (Figure 6).

**Table 3.** PLSR models built on the responses of the E-nose.

|  | Training Data | | Testing Data | |
|---|---|---|---|---|
|  | $R^2$ | RMSE | $R^2$ | RMSE |
| pH | 0.9489 | 0.0355 | 0.9137 | 0.0515 |
| COD | 0.9888 | 3.8406 | 0.9720 | 8.5404 |
| TN | 0.9486 | 0.7418 | 0.9447 | 0.9426 |
| TP | 0.9658 | 0.0458 | 0.9008 | 0.0789 |

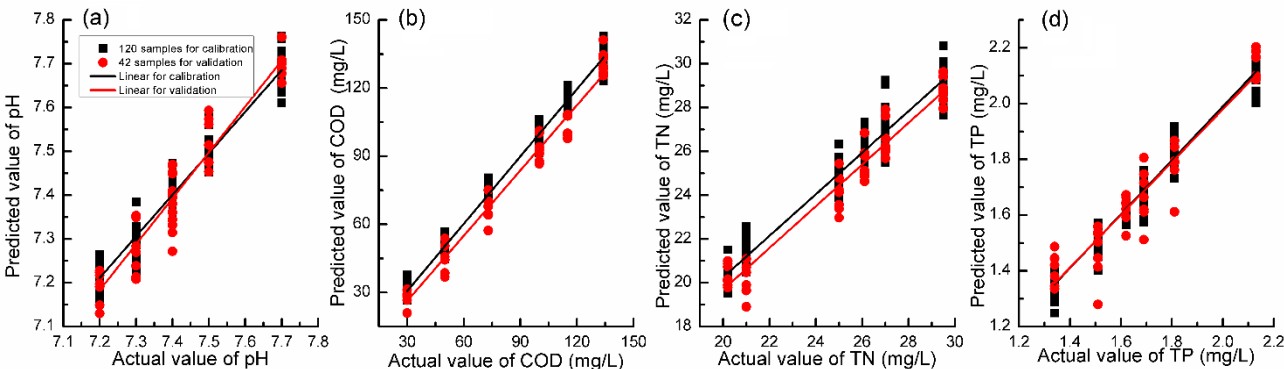

**Figure 6.** Prediction of river parameters using PLSR models based on E-nose signals: (**a**–**d**) are for the prediction of pH, COD, TN, and TP respectively based on E-nose.

## 4. Conclusions

Black-odor rivers are a severe problem that can affect the living conditions of residents and the functioning of local economies and ecosystems. Thus, there is an urgent need for an inexpensive, fast, and easy-to-use detection method for monitoring the conditions of black-odor rivers. Herein we presented a comprehensive and systematic experimental analysis of black-odor river water samples based on E-nose detection, which was applied as an alternative method to traditional monitoring technologies. Our conclusions are as follows:

(1) The values of pH, COD, and TN and TP contents of the Yueliang River obtained by chemical detection methods showed no correlation with each other according to Tukey's HSD test. The data distribution showed no significant pattern along the stretch of the Yueliang River sampled in this study.

(2) Correlations among E-nose sensor signals differed according to Pearson correlation matrix analysis, which means that information obtained from different sensors overlapped. ANOVA-PLSR results indicated that the E-nose sensors are cross-sensitive to specific compounds but fail to show relationships with all quality characteristics.

(3) Based on LDA, the E-nose system recognized the samples with 100% accuracy in the original data and cross-validation procedure. In addition, we used the PLSR model to reduce the risk of multicollinearity in the 10 E-nose sensors. For water quality parameter predictions, the coefficients between the actual water quality parameters (pH, COD, and TN and TP contents) and the predicted values were very high ($R^2 > 0.90$) in both the training and testing sets. This means that the E-nose technology successfully predicted the black-odor river water parameters using headspace gas measurements.

This is the first study of black-odor river water using an E-nose system to the best of our knowledge. This work illustrates that E-nose technology can be a fast, easy-to-build, and cost-effective detection system for black-odor river monitoring. Future work should focus on in situ E-nose monitoring of black-odor river water using a wireless network, which will result in online detection.

**Author Contributions:** Conceptualization, S.Q.; methodology; software, S.Q.; validation, J.H. and W.H.; formal analysis, Z.K.; investigation, P.H.; resources, P.H.; data curation, J.H.; writing—original draft preparation, S.Q.; writing—review and editing, S.Q. and J.H.; visualization, P.H.; supervision, P.H.; project administration, P.H.; funding acquisition, S.Q. and P.H. All authors have read and agreed to the published version of the manuscript.

**Funding:** Zhejiang Provincial National Science Foundation of China (LQ19D030002); National Key Re-search and Development Program-International Cooperation Project (2019YFE0124600); the Key Laboratory of Drinking Water Science and Technology, Research Center for Eco-Environmental Science, Chinese Academy of Sciences (19K01KLDWST).

**Institutional Review Board Statement:** Not applicable.

**Informed Consent Statement:** Not applicable.

**Data Availability Statement:** Not applicable.

**Acknowledgments:** The authors acknowledge the financial support of the Zhejiang Provincial National Science Foundation of China (LQ19D030002), the National Key Research and Development Program-International Cooperation Project (2019YFE0124600), and the Key Laboratory of Drinking Water Science and Technology, Research Center for Eco-Environmental Science, Chinese Academy of Sciences (19K01KLDWST).

**Conflicts of Interest:** The authors declare no conflict of interest.

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
