# Peer review of "The Monitoring of Black-Odor River by Electronic Nose with Chemometrics for pH, COD, TN, and TP"

_chemosensors, doi:10.3390/chemosensors9070168_

Round 1

Reviewer 1 Report

The manuscript under appreciation is about the application of an electronic nose (E-nose) for recognizing black-odor river samples and predict the quality parameters.

The manuscript is interesting and provides novelty and the results are important for future application.

The following comments are to be taken into account by the authors:

Table 2. Regarding quantification results and the relative standard deviation, the number of significant figures is indicative of the repeatability of the measurement, please report how many times did you performed the analysis and correct accordingly (triplicates?).

In order to apply ANOVA and LDA the data must be normalized and the Box M test (Equivalence of Covariance Matrices) must be non-significant. Did the authors follow these requirements? Please add this information in the manuscript

The LDA model must be validated. You must provide validation data (cross-validation or external validation) for the stepwise LDA.

Which software did the authors use to perform Pearson correlation? It’s not clearly reported.

In the PLSR the Root Mean Square Error of Prediction (RMSEP) values must also be reported (Table 3) in order to evaluate the performance of the model.

Author Response

Please see the attachment, and point-by-point responses have been provided in the attachment.

Reviewer 2 Report

General comments: this article presents an E-nose to detect off-flavour´s patterns in a black-odor river. They try to correlate with physicochemical parameters. But the election of these ones is not well justified in the introduction. The use of these parameters is neither mentioned in the article title.

Title: The title should include something related to physicochemical parameters

Abstract

Lines

28                   E-nose dispositive is really inexpensive? The authors should compare its price with physico-

chemical equipments applied to measure COD, TN, TP and pH.

31                  This sentence is inappropriate as there were no Tukey correlation between e-nose results and physico-chemical parameters.

  1. Introduction

Lines

63-4              Why these 4 parameters are the most important physico-chemical properties to measure water quality in rivers and lakes? The authors should use parameters most included in Water Quality Indexes to indicate water contamination.

  1. Materials and methods

The authors should report a contamination scale suitable to compare data obtained from e-nose and from physico-chemical properties.

Lines

103                One missprint was detected in this sentence.

Tables:

In Table 1, rows should be better distinguished between them.

Reviewer 3 Report

The authors reported the monitoring of black-odor river by E-nose technology. Qualitative recognition analysis and quantitative parameter prediction of samples have been performed. Nevertheless, there are many errors in the present manuscript. The methods of signal extraction and analysis, used by authors, are well-known to the researchers. The authors are recommended to extract more meaningful findings from experiments.

The additional comments are listed as following:

  1. The authors built the models through only six sampling sites, which is not enough. Besides, the testing data should be different from the training data of the six sampling sites to verify the performance of models.
  2. Line 100 on p. 3, "119.7°N" - no space

         Line 105 on p. 3, "20°C ± 0.5°C" - no space

         Table 2, "7.7±0.3 a ··· " - no space

  1. Line 141 on p. 4, "sample's" should be replaced by "samples".
  2. Line 252 on p. 6, "Fig. 1b shows ···" – Fig. 1b is disappeared.
  3. Y-axis of Figure 2 - "G/Go" should be replaced by "G/G0".
  4. The x-axis coordinates of Figure 4b should be parallel to the title.
  5. In Figure 6, "(a)", "(b)", "(c)", "(d)" should be parallel.

Round 2

Reviewer 1 Report

The authors have addressed all the issues. The manuscript is suitable for publication in Chemosensors.

Reviewer 3 Report

Some formats and details need further improvement.

Examples:

The height of each row in Table 1 is inconsistent.

Table 1 and line 147 are too close.

The font on lines 158 - 164 may be wrong. etc.....